# Polyethylene Glycol Priming Enhances the Seed Germination and Seedling Growth of *Scutellaria baicalensis* Georgi under Salt Stress

**DOI:** 10.3390/plants13050565

**Published:** 2024-02-20

**Authors:** Renjie Wang, Chenxuan Li, Li Zeng, Ligong Liu, Jiayi Xi, Jun Li

**Affiliations:** 1College of Agronomy, Qingdao Agricultural University, Qingdao 266109, China; wangrenjiex@163.com (R.W.); zengdali1996@163.com (L.Z.); xijiayi@163.com (J.X.); 2School of Traditional Chinese Medicine, Capital Medical University, Beijing 100069, China; chenxumari@163.com; 3National Engineering Research Center for Vegetables, Beijing Vegetable Research Center, Beijing Academy of Agriculture and Forestry Sciences, Beijing 100097, China

**Keywords:** *Scutellaria baicalensis* Georgi, polyethylene glycol (PEG), seed priming, salt tolerance

## Abstract

Seed priming has become a practical pre-sowing strategy to deal with abiotic stresses. This study aims to explore the effects of polyethylene glycol (PEG) priming on seed germination and seedling growth of *Scutellaria baicalensis* Georgi under salt stress. Regardless of seed priming, salt stress significantly inhibited the seed germination and seedling growth of *S. baicalensis*. PEG priming significantly alleviates the inhibitory effects of salt stress on seed germination and seedling growth when compared to non-priming and water priming. Among all treatments, PEG priming exhibited the highest germination rate, germination potential, seed vigor index, fresh weight, dry weight, and plant length; the highest contents of proline, soluble sugar, and soluble protein; the highest K^+^/Na^+^ ratio and relative water content; the highest antioxidant activities and contents; but the lowest H_2_O_2_, malondialdehyde (MDA) content, and relative electrical conductivity in response to salt stress. In addition, PEG priming had the highest transcript levels of antioxidant-related genes among all treatments under NaCl stress. Taken together, the results demonstrated that seed priming with PEG could be recommended as an effective practice to enhance the germination and early seedling growth of *S. baicalensis* under saline conditions.

## 1. Introduction

Soil salinization is an important environmental factor for crop planting, as it not only decreases planting area but also reduces the yield and quality of crops [1]. Globally, around one billion hectares of land are in high salinity, and the salinization of soil is becoming more and more serious during these decades, which is due to climate change, improper irrigation measures, and so on [1,2]. In China, about 1.3 × 10^7^ hectares of saline soils are potential agricultural land, accounting for about 10% of the total cultivated land area [3]. The fair use of these saline soils can increase cultivated land area effectively, meeting the diverse demands of the growing population.

The mechanisms of salt-induced damage in plants are well documented, including osmotic stress, oxidation stress, and ion toxicity. These deleterious effects ultimately lead to reduced water and nutrient absorption by plants [4,5]. To deal with the unfavorable stresses, plants have evolved a variety of adaptive mechanisms, such as up-regulating osmolyte contents, antioxidant systems, and ion homeostasis or regionalization. Notably, the accumulation of osmolytes, such as proline content, helps to improve water uptake, and the activation of antioxidant enzymes is beneficial to reduce ROS-induced oxidative damage [6,7].

Seed germination has been considered the most vulnerable stage in a plant’s life cycle, because salt stress can severely hinder seed sprouting and subsequent seedling establishment [8]. So, rapid seed germination and seedling establishment are vital for plants to survive in a saline environment [9]. To tackle this challenge, two methods have proven effective in agricultural practices: highly salt-tolerant varieties and the direct application of exogenous chemicals [10]. Seed priming is a pre-sowing treatment measure that involves initially controlling the water absorption of seeds and finally re-drying them [11]. And mounting evidence has shown that primed seeds are stronger than unprimed seeds in abiotic stresses [12,13,14]. Some chemical agents, such as phytohormones, reactive chemical species (H_2_O_2_ and NO), osmolytes, and nanoparticles, act in seed priming [15,16]. And the agent type, concentration, and treatment time all influence the effects of treatments [11]. Therefore, it is necessary to choose an optimal agent and the concentration for a certain crop in seed priming. Benadjaoud et al. [14] reported that 4% potassium chloride used in butterfly-lavender seed priming dramatically mitigated the negative impacts of salt or drought stress on germination.

*Scutellaria baicalensis* Georgi is a popular medicinal plant that is widely cultivated in China. Its dry root has been used as a kind of primary medicinal component to treat many diseases, such as hepatitis, dysentery, and epilepsy, for more than 2000 years [17,18]. Recent studies demonstrate that the active ingredients from the root of *S. baicalensis* exhibit preventive and therapeutic properties against a series of inflammatory responses triggered by severe acute respiratory syndrome coronavirus 2 (SARS-CoV-2), the causative agent of COVID-19 [19]. Currently, due to rapidly increasing demand, cultivated *S. baicalensis* has surpassed its wild counterpart as the primary market supply. However, soil salinization has seriously endangered the planting area and yield of *S. baicalensis*. A high level of seed germination and seedling establishment is essential for *S. baicalensis* to survive in a saline habitat. This study aims to explore the effects of polyethylene glycol (PEG) priming on seed germination and seedling growth of *S. baicalensis* under salt stress. It provides an empirical foundation for the practical production and selection of salt-tolerant varieties of *S. baicalensis*.

## 2. Results

### 2.1. PEG Priming Significantly Alleviates the Germination Inhibition of S. baicalensis Seeds under Salt Stress

Regardless of seed priming, salt stress significantly inhibited the germination of *S. baicalensis* seeds. Compared to the non-priming control, priming treatments alleviated the negative effect of salt stress on seed germination (Figure 1). The germination potential (GP), germination rate (GR), and seed vigor index (SVI) under W1 treatment were increased by 14.3%, 25.9%, and 64.2%, respectively, compared to that under N1 treatment. The GP, GR, and SVI under P1 treatment were enhanced by 51.4%, 81.5%, and 155.7%, respectively, over N1 treatment. 

### 2.2. PEG Priming Significantly Enhanced the Growth of S. baicalensis Seedlings under Salt Stress

Regardless of seed priming, salt stress significantly impaired the growth of *S. baicalensis* seedlings. Compared to the non-priming control, priming treatments significantly mitigated the inhibitory effect of salt stress on the growth of seedlings (Figure 2a). The plant length (PL) of *S. baicalensis* seedlings under salt stress was significantly decreased. Compared to the N0 treatment, the PL under N1, W1, and P1 treatments was decreased by 76.5%, 58.8%, and 39.7%, respectively. Upon exposure to salt stress, the seedlings under priming treatments showed higher PL than those in the non-priming control. The PLs under W1 and P1 treatments were 1.75- and 2.56 times that under N1 treatment, respectively (Figure 2b). 

The fresh weight (FW) and dry weight (DW) of seedlings under salt stress were significantly decreased. Compared to the N0 treatment, N1, W1, and P1 treatments resulted in significant decreases in the FW (49.5%, 29.9%, and 19.6%) and DW (42.8%, 18.6%, and 11.3%), respectively. Compared to the non-priming control, priming treatments significantly enhanced the FW and DW under salt stress. W1 and P1 treatments caused significant enhancements in the FW (38.8% and 59.2%, respectively) and DW (42.3% and 54.9%) compared to the N1 treatment (Figure 2c,d). 

### 2.3. Effect of PEG Priming on Osmolyte Content and Relative Water Content of S. baicalensis Shoots under Salt Stress

Salt stress significantly enhanced the contents of proline and soluble protein while diminishing the soluble sugar concentration and relative water content (RWC) in *S. baicalensis* shoots compared to non-salt treatment (Figure 3). Upon exposure to salt stress, seedlings in the PEG priming treatment exhibited higher contents of proline, soluble protein, soluble sugar, and RWC than in the non-priming control. The contents of proline, soluble protein, soluble sugar, and RWC under P1 treatment were increased by 66.3%, 8.6%, 5.4%, and 13.9%, respectively, over the N1 treatment.

### 2.4. Effect of PEG Priming on Lipid Peroxidation and Membrane Permeability of S. baicalensis Shoots under Salt Stress

In the non-priming treatments, salt stress significantly increased the H_2_O_2_ content, MDA content, and REC in the shoots of *S. baicalensis* (Figure 4a–c). The H_2_O_2_ content, MDA content, and REC under N1 treatment were significantly enhanced by 64.1%, 55.0%, and 115.9%, respectively, over the N0 treatment. Both water and PEG priming significantly reduced the H_2_O_2_ content, MDA content, and REC, with PEG priming exhibiting a larger decline than water priming. Compared to the N1 treatment, the W1 and P1 treatments led to significant decreases in the H_2_O_2_ content (6.0% and 42.4%, respectively), MDA (20.8% and 29.7%), and REC (28.8% and 33.7%).

### 2.5. Effect of PEG Priming on Antioxidant Systems of S. baicalensis Shoots under Salt Stress

Upon exposure to salt stress, PEG priming significantly increased the activities of enzymatic antioxidants and the content of non-enzymatic antioxidants in the shoots of *S. baicalensis*, compared to the non-priming control (Figure 4d–f and Figure 5). The activities of SOD, POD, and CAT under P1 treatment were enhanced by 46.4%, 65.9%, and 59.6%, respectively, over the N1 treatment. The ascorbic acid (AsA) content, glutathione (GSH) content, and GSH/GSSG ratio under P1 treatment were respectively enhanced by 63.2%, 10.5%, and 117.9%, but the GSSG content decreased by 50.1% compared to the N1 treatment. 

### 2.6. Effect of PEG Priming on Contents of Sodium and Potassium Ions in S. baicalensis Shoots under Salt Stress

Compared to the non-salt treatment, salt stress significantly enhanced the Na^+^ content but decreased the K^+^ content in the shoots of *S. baicalensis* (Figure 6). The contents of Na^+^ under N1, W1, and P1 treatments were respectively increased by 273.9%, 188.5%, and 182.2%, while the contents of K^+^ were respectively decreased by 32.6%, 25.1%, and 22.2%, over the N0 treatment. Compared to the non-priming treatment, priming treatment significantly decreased the Na^+^ content but enhanced the K^+^ content under salt stress. The Na^+^ contents under W1 and P1 treatments were respectively decreased by 22.9% and 24.5%, while the K^+^ contents were respectively increased by 11.2% and 15.4%, over the N1 treatment. Furthermore, PEG priming and water priming showed a higher K^+^/Na^+^ ratio than in the non-priming treatment under salt stress. The K^+^/Na^+^ ratios in W1 and P1 were enhanced by 44.1% and 52.9%, respectively, over N1 treatment.

### 2.7. Effect of PEG Priming on Transcript Levels of Antioxidant-Related Genes in S. baicalensis Shoots under Salt Stress

Compared to the non-priming control, priming treatments significantly increased the transcript levels of antioxidant-related genes, and PEG priming demonstrated greater effectiveness than water priming under salt stress (Figure 7). The transcript levels of *SbPOD1*, *SbPOD2*, *SbPOD3*, *SbSOD1*, *SbSOD2*, and *SbAPX* under P1 treatment were significantly enhanced by 57.1%, 103.2%, 87.9%, 90.5%, 62.6%, and 44.9%, respectively, over N1 treatment.

## 3. Discussion

Abiotic stresses, such as salt and drought, are the key ecological factors that limit the growth and quality of traditional Chinese medicine [20]. Seed priming is considered a promising technique for increasing seed germination and seedling growth under challenging environmental conditions. This technique has been proven effective in a variety of plant species, including food crops, certain vegetables, and flowers [9,15,21]. Polyethylene glycol (PEG) is an osmotic agent commonly used for simulating the conditions of drought stress in plants [22]. PEG priming efficiently improved the germination rate and seedling growth of barley in response to drought stress [23]. In the presented study, salt stress significantly inhibited the seed germination and seedling growth of *S. baicalensis*, regardless of whether the seeds were primed or not. This is in agreement with the other studies that demonstrated negative effects of salt stress on plant growth [14,24]. PEG priming showed significantly enhanced seed germination rate and seedling growth of *S. baicalensis* under salt stress, certificating the positive role of seed priming in plant response to environmental stress. 

Although it is a highly energy-consuming process, the rapid biosynthesis of certain compatible solutes, such as soluble sugar and proline, remains an efficient strategy for plants to cope with osmotic stress [25]. These compatible solutes play important roles in the enhancement of salt and drought tolerance in many plant species [26,27]. Parallel to the observations, our studies revealed that the contents of proline and soluble protein in *S. baicalensis* seedlings were significantly increased in response to salt stress and seed priming, with either water or PEG further enhancing their contents. Soluble sugar, an important osmolyte, does not always show an increasing trend in response to salt stress. The possible reason is that the decomposition of these carbohydrates is conducive to providing sufficient carbon backbone and energy supply for the synthesis of other osmotic protectants containing nitrogen compounds to cope with the deepening of salt stress [28]. Wang et al. [29] reported that salt stress resulted in the decrease in soluble sugar content of *G. uralensis* seedlings. RWC is an important indicator to assess environmental stress tolerance. Vazayef et al. [30] reported that a high level of salinity resulted in a significantly reduced RWC in leaves of *Brassica napus*. In this study, salt stress significantly decreased the soluble sugar content and RWC in *S. baicalensis* shoots but seed priming with PEG promoted the accumulation of soluble sugar and RWC, which agreed with previous studies reported by Wang et al. [29] and Vazayef et al. [30].

Ion homeostasis plays a vital role in plant salt tolerance. An excess accumulation of sodium ions (Na^+^) under salt stress would disturb the balance between Na^+^ and other ions, such as potassium ions (K^+^) and calcium ions, (Ca^2+^), and eventually lead to ion toxicity [30]. In this study, we also found a significantly enhanced Na^+^ content and a significantly reduced K^+^ content in *S. baicalensis* shoots upon exposure to salt stress. Potassium, an essential macronutrient, plays an important role in regulating the activity of various enzymes during plant growth and development. A high ratio of K^+^/Na^+^ is required for plants to minimize Na^+^ toxicity and maintain ion homeostasis in saline environments [31]. In the presented study, the ratio of K^+^/Na^+^ in seedlings with seed priming was significantly higher than that in seedlings without any priming treatment under salt stress. It was helpful for *S. baicalensis* seedlings to maintain the ion homeostasis and minimize Na^+^ toxicity caused by salt stress.

Besides osmotic stress and ion toxicity, salt stress generates excess reactive oxygen species (ROS), such as H_2_O_2_ in plants, and easily leads to oxidative stress [32]. The changes in MDA levels and REC can be served as important markers for evaluating the degree of oxidative damage of cell membranes [33]. So, in the experiments presented here, the contents of H_2_O_2_, MDA, and REC were significantly enhanced in the seedlings treated by salt stress, suggesting that salt stress markedly increased lipid peroxidation. However, the degree of lipid peroxidation was greatly alleviated in the seedlings with seed priming, and PEG priming was more efficient than water priming. In plants, two sets of antioxidant systems are developed to minimize the adverse impacts from ROS: one is enzymatic antioxidants, such as SOD and CAT, and the other is non-enzymatic antioxidants, such as GSH and AsA [7,34]. Here, we found that salt stress mildly increased SOD activity and GSH content but significantly enhanced POD, CAT activities, and AsA content. The positive effects of PEG priming on the antioxidant systems of *S. baicalensis* shoots were stronger than those in non-priming and water-priming under salt stress. This is in line with the previous results that seed priming conferred drought tolerance to Chinese cabbage seedlings by up-regulating the activities of the antioxidant enzymes [12].

In the *S. baicalensis* genome, we found six genes encoding antioxidant enzymes (*SbSOD1*, *SbSOD2*, *SbPOD1*, *SbPOD2*, *SbPOD3*, and *SbAPX*) by exploring the National Centre for Biotechnology Information (NCBI) website (https://www.ncbi.nlm.nih.gov). In the presented experiment, we found that salt stress mildly down-regulated the expression of *SbPOD1* and significantly decreased the expression of *SbSOD1* but significantly increased the expression of the others (*SbSOD2*, *SbPOD1*, *SbPOD2*, *SbPOD3*, and *SbAPX*). This suggested that the expression pattern of these genes varied in response to salt stress. In addition, the transcript levels of *SbSOD1*, *SbSOD2*, *SbPOD1*, *SbPOD2*, *SbPOD3*, and *SbAPX* were found to be significantly enhanced in seedlings treated with PEG priming, indicating that PEG priming greatly up-regulated these genes’ expression levels. This can explain the highest activities of SOD, POD, and CAT in *S. baicalensis* shoots treated with PEG priming. 

## 4. Materials and Methods

### 4.1. Plant Materials and Priming

Seeds of *S. baicalensis* Georgi were collected from its native producing area in Longxi County, Gansu Province, China. A total of 360 healthy and uniform seeds underwent surface sterilization with a 2% sodium hypochlorite solution for 10 min. Following this, they were rinsed four or five times with distilled water. The seeds were divided equally into three groups, two of which were soaked with distilled water or 20% PEG-6000 at room temperature for 12 h, respectively. After that, these seeds were washed thoroughly with distilled water and air-dried at room temperature for two days back to their initial moisture content. Another group without soaking served as the unprimed control. Both primed and unprimed seeds were set to germinate in Petri dishes, each lined with two layers of filter paper (Whatman #1, 60 seeds per dish). The filter paper was dampened with 2 mL of distilled water or 75 mM of NaCl solution.

There were six seed treatments in total: (1) non-priming without salt treatment (N0); (2) non-priming and cultivated with 75 mM of NaCl solution (N1); (3) water priming without salt treatment (W0); (4) water priming and cultivated with 75 mM of NaCl solution (W1); (5) PEG priming without salt treatment (P0); (6) PEG priming and cultivated with 75 mM of NaCl solution (P1). These seeds were placed in a germinator at 25 °C under 16 h/8 h of light/dark condition. The seeds per dish were sprayed with 1.0 mL of water or 75 mM of NaCl solution every 2 days. At least three independent biological replicates were used for each treatment.

### 4.2. Germination Test

The germination rate (GR), germination potential (GP), and seedling vigor index (SVI) were measured according to the methods described by Yan [12]. Seeds with 1 mm long radicle protruded through the seed coat were considered germinated. The GP was investigated on the 5th day and GR was on the 12th day.

### 4.3. Measurements of Growth Parameters and Sampling

After the germination test, three seedlings from each treatment were randomly collected and photographed. For each treatment, ten seedlings were selected to measure the plant length (PL), fresh weight (FW), and dry weight (DW). Relative water content (RWC) in shoots was determined as described by Li et al. [10]. PL was the distance from taproot tip to shoot unbent. The other seedlings were separated into shoots and roots, and the shoots were for later determining the physiological index and performing the molecular detection.

### 4.4. Assay of Physiological Indicators and Ion Contents

The MDA content, soluble protein content, and soluble sugar content were quantified following the methodologies delineated by Yusuf et al. [35], Aminian et al. [36], and Hackmann et al. [37], respectively. REC was determined by the conductance method. Proline content was measured according to the method described by Benitez et al. [38]. The contents of AsA, GSH, and GSSG (GSH, oxidized) were determined using the corresponding assay kit (Beijing Solarbio Science & Technology Co., Ltd., Beijing, China). The H_2_O_2_ content was assayed following the methods in Martinez-Gutierrez et al. [39]. The activities of SOD, POD, and CAT were measured following the methods described by Usluoglu et al. [40]. The contents of Na^+^ and K^+^ were determined by flame photometry (ICP-OES, Optima 8000, PerkinElmer Instruments Co., Ltd., Waltham, MA, USA) as described by Williams and Twine [41].

### 4.5. qRT-PCR Analysis

Total RNA was isolated with an RNA extraction kit (Tiangen, Beijing, China) and then reverse transcribed into cDNA using a PrimeScript RT Reagent Kit (Tiangen, Beijing, China). Using *SbActin* as an internal reference, qRT-PCR was conducted to assess the expression levels of genes associated with antioxidant enzymes via specific primers (Appendix A). The antioxidant-enzyme-related genes included *SbSOD1* (HQ395746), *SbSOD2* (HQ395747), *SbPOD1* (AB024437), *SbPOD2* (AB024438), *SbPOD3* (AB024439), and *SbAPX* (HQ395752). The relative expression level was calculated using the 2^–ΔΔt^ method [42]. All data are presented as the mean ± SD after normalization.

### 4.6. Statistical Analysis

Analysis-of-variance (ANOVA) procedures were performed with the SPSS 22.0 software (IBM, Armonk, NY, USA). Using the SAS program, the means were distinguished based on the least-significant-difference (LSD) test at a significance level of 0.05.

## 5. Conclusions

In general, PEG priming effectively alleviates the adverse effect of salt stress on the seed germination and seedling growth of *S. baicalensis* (Figure 8). This was due to PEG priming enhancing the accumulation of osmolytes, such as proline content, and improving the K^+^/Na^+^ ratio to tune down osmotic stress and ion toxicity; PEG priming also increased the AsA content and enzymatic activities through up-regulating the expression levels of antioxidant-related genes. In addition, PEG priming reduced the contents of H_2_O_2_ and MDA to minimize the oxidative stresses of *S. baicalensis* seedlings. 

## Figures and Tables

**Figure 1 plants-13-00565-f001:**
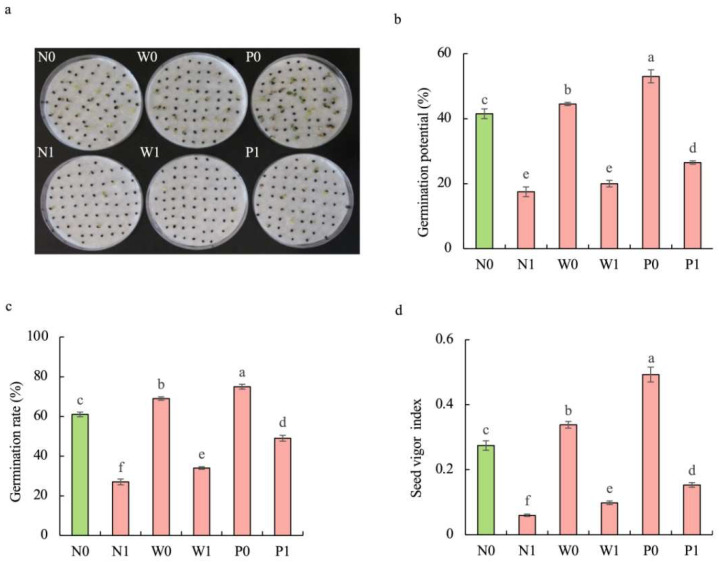
Effects of PEG priming on the germination of *S. baicalensis* seeds in response to salt stress. (**a**) Phenotype; (**b**) germination potential; (**c**) germination rate; (**d**) seed vigor index. Data represent mean ± SD (*n* = 3). Bars annotated by different letters indicate statistically significant differences between treatments, as determined by the LSD test at a significance level of *p* < 0.05. N0: non-priming without salt treatment. N1: non-priming and cultivated with 75 mM of NaCl solution. W0: water priming without salt treatment. W1: water priming and cultivated with 75 mM of NaCl solution. P0: PEG priming without salt treatment. P1: PEG priming and cultivated with 75 mM of NaCl solution.

**Figure 2 plants-13-00565-f002:**
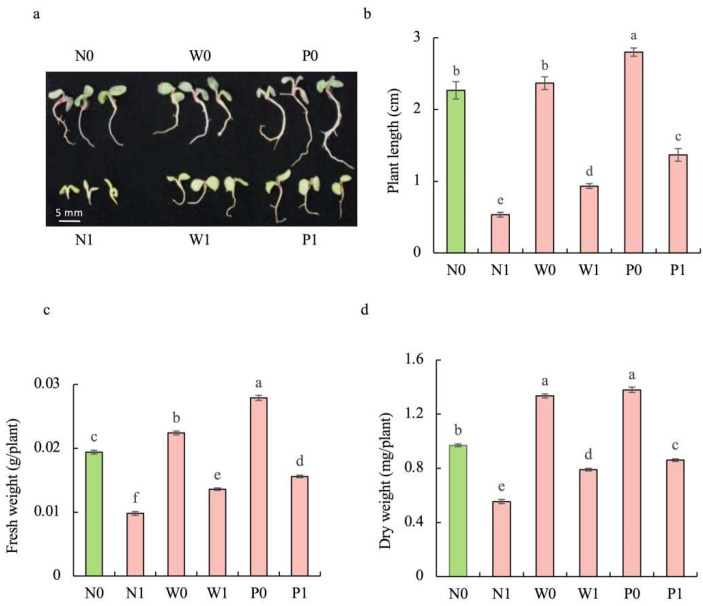
Effects of PEG priming on the growth of *S. baicalensis* seedling in response to salt stress. (**a**) Phenotype; (**b**) plant length; (**c**) fresh weight; (**d**) dry weight. Data represent mean ± SD (*n* = 3). Bars annotated by different letters indicate statistically significant differences between treatments, as determined by the LSD test at a significance level of *p* < 0.05. N0: non-priming without salt treatment. N1: non-priming and cultivated with 75 mM of NaCl solution. W0: water priming without salt treatment. W1: water priming and cultivated with 75 mM of NaCl solution. P0: PEG priming without salt treatment. P1: PEG priming and cultivated with 75 mM of NaCl solution.

**Figure 3 plants-13-00565-f003:**
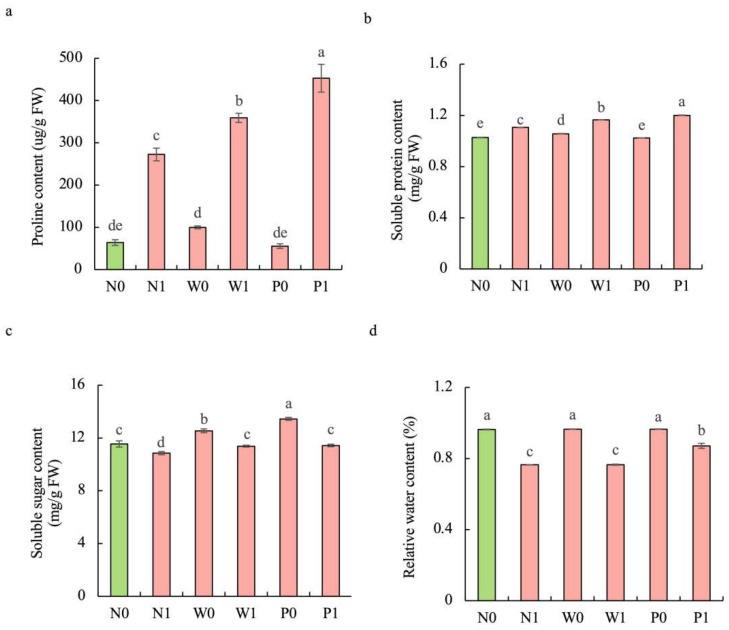
Effects of PEG priming on the osmolyte content and relative water content (RWC) of *S. baicalensis* shoots in response to salt stress. (**a**) Proline content; (**b**) soluble protein content; (**c**) soluble sugar content; (**d**) RWC. Data represent mean ± SD (*n* = 3). Bars annotated by different letters indicate statistically significant differences between treatments, as determined by the LSD test at a significance level of *p* < 0.05. N0: non-priming without salt treatment. N1: non-priming and cultivated with 75 mM of NaCl solution. W0: water priming without salt treatment. W1: water priming and cultivated with 75 mM of NaCl solution. P0: PEG priming without salt treatment. P1: PEG priming and cultivated with 75 mM of NaCl solution.

**Figure 4 plants-13-00565-f004:**
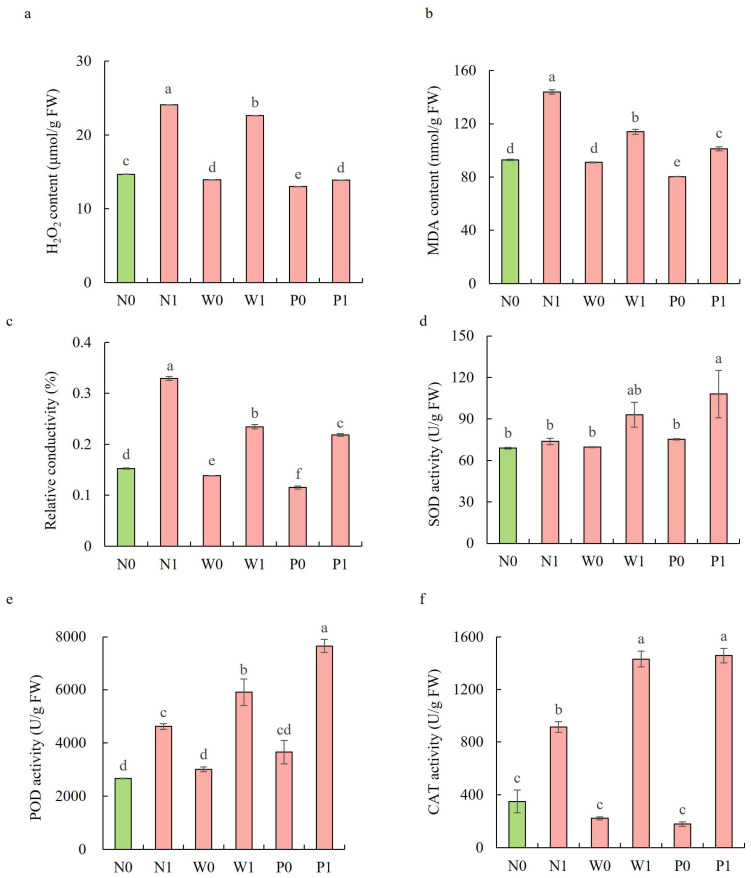
Effects of PEG priming on lipid peroxidation, membrane permeability, and enzymatic antioxidant activity of *S. baicalensis* shoots in response to salt stress. (**a**) H_2_O_2_ content; (**b**) MDA content; (**c**) relative electrical conductivity (REC); (**d**) SOD activity; (**e**) POD activity; (**f**) CAT activity. Data represent mean ± SD (*n* = 3). Bars annotated by different letters indicate statistically significant differences between treatments, as determined by the LSD test at a significance level of *p* < 0.05. N0: non-priming without salt treatment. N1: non-priming and cultivated with 75 mM of NaCl solution. W0: water priming without salt treatment. W1: water priming and cultivated with 75 mM of NaCl solution. P0: PEG priming without salt treatment. P1: PEG priming and cultivated with 75 mM of NaCl solution.

**Figure 5 plants-13-00565-f005:**
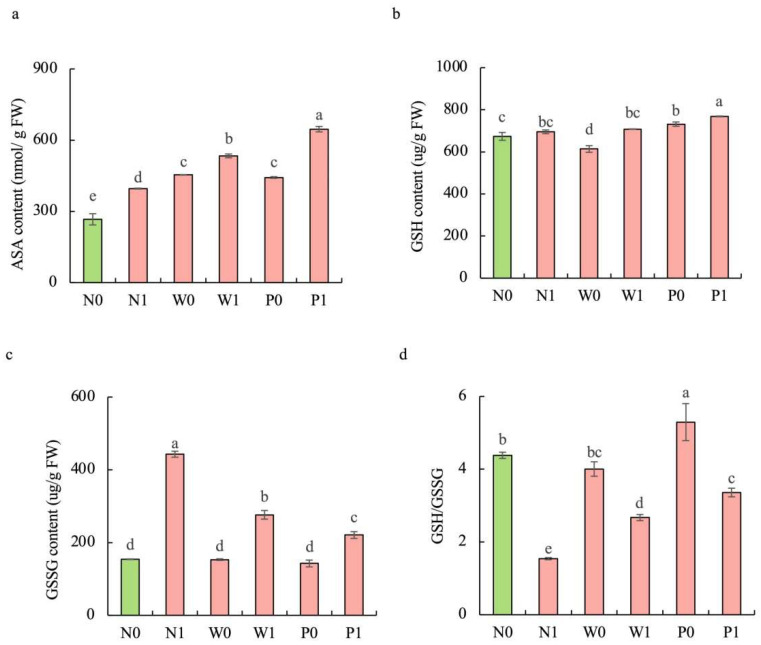
Effects of PEG priming on the non-enzymatic antioxidant content of *S. baicalensis* shoots in response to salt stress. (**a**) AsA content; (**b**) GSH content; (**c**) GSSG content; (**d**) GSH/GSSG ratio. Data represent mean ± SD (*n* = 3). Bars annotated by different letters indicate statistically significant differences between treatments, as determined by the LSD test at a significance level of *p* < 0.05. N0: non-priming without salt treatment. N1: non-priming and cultivated with 75 mM of NaCl solution. W0: water priming without salt treatment. W1: water priming and cultivated with 75 mM of NaCl solution. P0: PEG priming without salt treatment. P1: PEG priming and cultivated with 75 mM of NaCl solution.

**Figure 6 plants-13-00565-f006:**
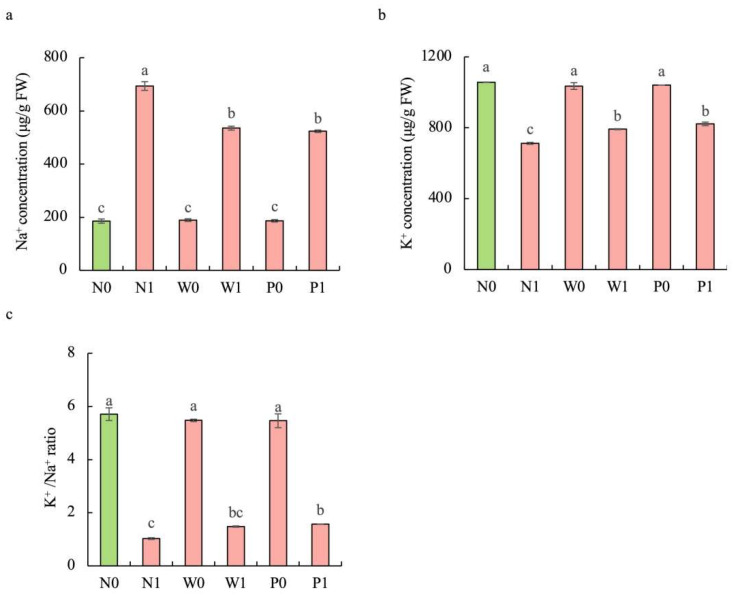
Effects of PEG priming on the Na^+^ content, K^+^ content, and K^+^/Na^+^ ratio of *S. baicalensis* shoots in response to salt stress. (**a**) Na^+^ content; (**b**) K^+^ content; (**c**) K^+^/Na^+^ ratio. Bars annotated by different letters indicate statistically significant differences between treatments, as determined by the LSD test at a significance level of *p* < 0.05. N0: non-priming without salt treatment. N1: non-priming and cultivated with 75 mM of NaCl solution. W0: water priming without salt treatment. W1: water priming and cultivated with 75 mM of NaCl solution. P0: PEG priming without salt treatment. P1: PEG priming and cultivated with 75 mM of NaCl solution.

**Figure 7 plants-13-00565-f007:**
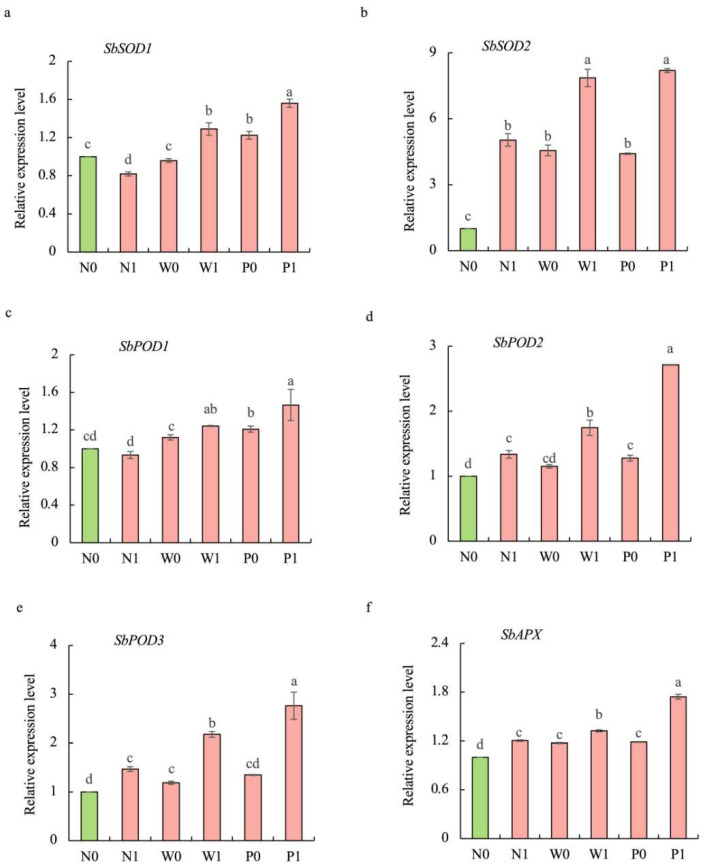
Effects of PEG priming on the transcription levels of antioxidant-related genes of *S. baicalensis* shoots in response to NaCl stress. (**a**) *SbSOD1*; (**b**) *SbSOD2*; (**c**) *SbPOD1*; (**d**) *SbPOD2*; (**e**) *SbPOD3*; (**f**) *SbAPX*. Data represent mean ± SD (*n* = 3). Bars annotated by different letters indicate statistically significant differences between treatments, as determined by the LSD test at a significance level of *p* < 0.05. N0: non-priming without salt treatment. N1: non-priming and cultivated with 75 mM of NaCl solution. W0: water priming without salt treatment. W1: water priming and cultivated with 75 mM of NaCl solution. P0: PEG priming without salt treatment. P1: PEG priming and cultivated with 75 mM of NaCl solution.

**Figure 8 plants-13-00565-f008:**
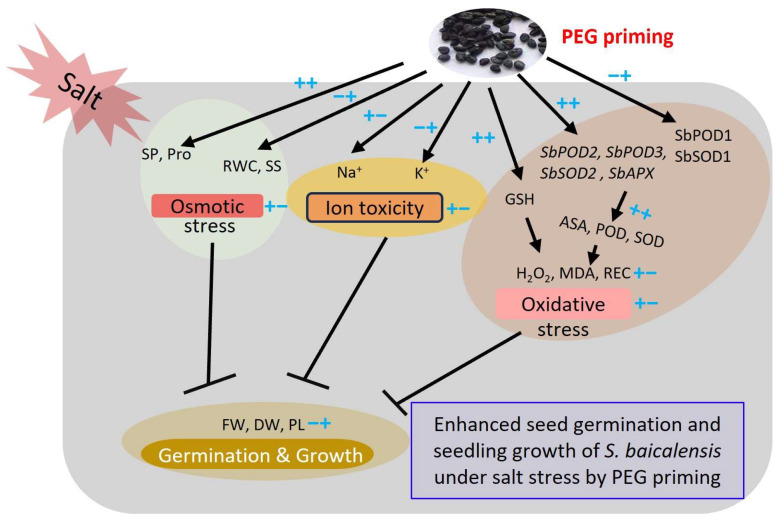
Working model of enhanced seed germination and seedling growth of *S. baicalensis* under salt stress induced by PEG priming. ++ or −− refers to the strong increase or decrease treated by both salt stress and PEG priming, respectively. +− indicates the initial increase by salt stress and subsequent decrease caused by PEG priming. −+ indicates the initial decrease by salt stress and subsequent increase by PEG priming. SOD, POD, CAT, PEG, Pro, RWC, GSH, AsA, REC, MDA, FW, DW, SP, SS, and PL are the abbreviations of superoxide dismutase, peroxidase, catalase, polyethylene glycol, proline, relative water content, glutathione, ascorbic acid, relative electrical conductivity, malondialdehyde, fresh weight, dry weight, soluble protein, soluble sugar, and plant length, respectively.

## Data Availability

Data are contained within the article and Appendix A.

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
