# Peer review of "Polyethylene Glycol Priming Enhances the Seed Germination and Seedling Growth of Scutellaria baicalensis Georgi under Salt Stress"

_plants, 2024, doi:10.3390/plants13050565_

Round 1

Reviewer 1 Report

Comments and Suggestions for Authors

There are lacking definitions for “N0, N1, W0, W1, P0, P1” in the main text and figure captions, which makes it impossible to understand the figures and the manuscript.

What do the letters over the bars in the figure stand for? Please provide the definitions.

Line 66, Delete “artificially”

Line 67, replace “supplanted” with “surpassed”.

Line100-105, these comparisons are very confusing for me.

Reviewer 2 Report

Comments and Suggestions for Authors

Dear Editor

Thank you very much for your invitation to review this manuscript (plants-2818805).

Seed Priming with Polyethylene Glycol Alleviates the Negative Effects of Salt Stress on Germination and Seedling Growth of Scutellaria baicalensis Georgi

·         In general, the idea is good and there are many comments highlighted in the pdf version, please see the pdf version and the following comments:

·         Lines 17-21 In the abstract: rewrite this phrase.

·         In the introduction section you should cite recent references.

·         Lines 85-87:   add explanation for NO, N1, W0, ...etc

·         Lines 107-109:   add explanation for NO, N1, W0, ...etc

·         In the discussion section such as Line 240 - It is preferable to cite recent references, you can see the following papers:

https://www.mdpi.com/2076-3921/10/3/398

https://www.mdpi.com/2311-7524/9/6/711

https://www.mdpi.com/2077-0472/12/12/2084

·         Line 287- revise the reference of RWC determination, it is not LI et al. method

Revise and rewrite the correct authors.

·         Line 360: follow the roles of the Journal in the references.

·         Line 413: delete , and add ; instead.

·         Line 417: delete , and add ; instead.

·         Please revise the references list according to the Journal roles.

·         See the comments in the pdf version

Best regards

Comments on the Quality of English Language

Minor editing of English language required

Reviewer 3 Report

Comments and Suggestions for Authors

In this study, the authors demonstrated seed priming with polyethylene glycol alleviates the negative effects of salt stress on germination and seedling growth of Scutellaria baicalensis Georgi, and this alleviation is more significant than water priming. My minor comments/suggestions are as follows:

1.     Line 279 “Germination potential (GP), germination rate (GR), germination index (GI) and seed- 279

1.ling vigor index (SVI) were measured according to the methods described by Yan”, however, there were no GI-associated results in this manuscript.

2.     Lines 115-117, “Upon exposure to salt stress, seedlings in PEG priming treatment exhibited higher contents of proline, soluble protein, soluble sugar and RWC than in non-priming control and water priming treatment”. This conclusion is not correct, in Figure 3c, the PEG priming has no difference from water priming in affecting soluble sugar.

3.     Line 164 “PEG priming showed a higher K+/Na+ ratio than non-priming and water-priming treatments under salt stress” This conclusion is not correct, in Figure 3c, the PEG priming has no difference from water priming in K+/Na+ ratio.

4.     Lines 132-133, “Relative to N1 treatment, the H2O2 content, MDA content and REC under W1 and P1 treatments were decreased by 6.0%, 20.8%, 28.8%, 42.4%, 29.7% and 33.7%, respectively” and lines 175-177 “The transcript levels of SbPOD1, SbPOD2, SbPOD3, SbSOD1, SbSOD2 and SbAPX under P1 treatment were significantly up-regulated, showing increases of 57.1%, 103.2%, 87.9%, 90.5%, 62.6% and 44.9%, respectively, over N1 treatment” these sentences need to be rewritten as so many data jumbled up, making it hard to read.

5.     Line 310, when mentioning the primers, it would be better to show the concentration, not only the volume.

6.     Mistakes in spelling: Line 98 “that” should be “than”; Line 315 “showed” should be “shown”.

7.     MDA and NCBI should be given their full name when first mentioned in the manuscript.

Round 2

Reviewer 1 Report

Comments and Suggestions for Authors

I’m ok with the revised.